# Likelihood of Preterm Birth in Patients After Antenatal Corticosteroid Administration in Relation to Diagnosis and Confounding Risk Factors: A Retrospective Cohort Study

**DOI:** 10.3390/healthcare13010087

**Published:** 2025-01-06

**Authors:** Jan Pauluschke-Fröhlich, Richard Berger, Harald Abele, Claudia F. Plappert, Joachim Graf

**Affiliations:** 1Department of Women’s Health, University Hospital Tübingen, 72076 Tubingen, Germany; jan.pauluschke-froehlich@med.uni-tuebingen.de (J.P.-F.); harald.abele@med.uni-tuebingen.de (H.A.); 2Department of Obstetrics and Gynecology, Hospital Marienhaus, 56564 Neuwied, Germany; richard.berger@marienhaus.de; 3Institute for Health Sciences, Department of Midwifery Science, University Hospital Tübingen, 72076 Tübingen, Germany; claudia.plappert@med.uni-tuebingen.de

**Keywords:** premature birth, antenatal corticosteroid administration, risk factors, moderation effects

## Abstract

*Background*: In the case of threatened preterm birth (PTB) before the 34th week of pregnancy, the application of antenatal corticosteroids (ACSs) for the maturation of the fetal lung is a standard procedure in perinatal medicine. Common diagnoses for ACS use in pregnancy are the preterm rupture of membranes (PPROMs), placental bleeding, premature labor, preeclampsia, oligohydramnios, amniotic infection syndrome (AIS), and cervical insufficiency. The aim of this study was to investigate whether the current diagnosis, which results in ACS, and the patient’s risk factors influence the risk of PTB events. *Methods*: The data of all affected women were extracted, who were hospitalized in 2016 due to a threatening PTB and administered corticosteroids in the German federal state Rhineland-Palatinate (*n* = 1544), so the study was conducted as a retrospective cohort trial. Frequency analyses, Friedman tests, Chi square tests, logistic regressions, Spearman correlation, and moderation analysis were performed to determine the Odds ratio (OR) for PTB in ACS patients in relation to diagnosis and risk factors. *Results*: Only 60% of all patients with PTB risk delivered prematurely, whereby patients with different diagnoses differ in terms of the PTB risk; the highest risk could be found in AIS (OR = 16.12) or preeclampsia (OR = 5.46). For prior PTB, stillbirth, or abortion, there is a moderation effect (based on the confounders), while multiple pregnancies influence the PTB risk irrespective of diagnosis (OR = 6.45). In the whole collective, the OR for PTB was 6.6 in relation to all pregnant women in Germany. *Conclusions*: A higher risk of PTB could be found in patients with a diagnosis of AIS, preeclampsia, as well as in multiple pregnancies. Prior PTB, stillbirth, or abortion act as a relevant confounder.

## 1. Introduction

With an incidence of around 8% of all births, preterm birth (PTB) belongs to the most frequent birth complications in Germany [1]. Although the PTB rate in Germany has been constant for many years, it is higher than in other European countries [2], which is why the identification and interaction of various risk factors is highly relevant [3]. PTB is defined as a birth before the 37 + 0 gestational week or before the 260th day of pregnancy [4]. Prematurity is one of the central problems of obstetrics, and it is considered the most important risk factor for perinatal mortality and morbidity [5,6]. The most common causes are ascending infections, hypoxic-ischemic impairments of the utero-placental unit, chronic stress, as well as fetal and uterine malformations [7]. Significant risk factors include a burdened obstetric anamnesis (adverse PTBs or late miscarriages), unfavorable socio-economic living conditions, pregnancy without a partner, unhealthy lifestyle (e.g., smoking), multiple pregnancies, and a low or high maternal age (<18 years and >35 years) [8,9]. In the case of threatened PTB before the 34th week of pregnancy, the application of antenatal corticosteroids (ACSs) for the maturation of the fetal lung is a standard procedure in perinatal medicine [10,11]. It is of relevance that not every patient with a threatening PTB and possibly corticosteroid administration goes on to have a PTB. So far, there is a lack of cohort studies with large collective groups which investigate how often PTB actually occurs in patients with a threatened PTB, depending on the respective causes for induction for ACS and defined risk factors for PTB. It remains unclear whether there is a special risk collective that benefits from the well-timed induction of ACS. This is also relevant because the administration of corticosteroids has side effects on the long-term development of the child; therefore, the indication for lung maturation must be made carefully [12,13]. Against this background, the World Association of Perinatal Medicine (WAPM) also recommends a restrictive use of ACS and calls for more research, particularly to more accurately predict which women are highly likely to experience PTB within the next seven days. The aim of the WAPM recommendation is to increase the timely administration of treatment and to avoid unnecessary or excessive use of ACS, which requires the precise identification of high-risk groups [13]. This is also relevant as the guidelines have not yet provided a standardized definition and stratification of when and with what probability a threatened PTB is to be expected. This is usually determined by clinicians when one of the diagnoses, listed below, is present, although the diagnosis of PPROM is also complicated by the lack of standardized diagnostic criteria [13].

The study analyzed a total of *n* = 1544 patients with the clinical diagnosis of “threatened PTB” who had received corticosteroids (ACSs) for drug-induced fetal lung maturation. The aim was to show how often within an ACS cohort the diagnoses of preterm premature rupture of membranes (PPROMs), placental bleeding, preterm labor, preeclampsia, oligohydramnios, amniotic infection syndrome (AIS), and cervical insufficiency are distributed throughout the cohort and to determine how often these patients have additional risk factors (age of the mother > 35, condition after PTB, stillbirth, or abortion in personal anamnesis, multiple pregnancy, gestational diabetes, hypertension, and nicotine abuse). Furthermore, we aim to investigate whether PTB shows dependencies between the diagnostic groups and the risk factors, and whether the risk factors in particular lead to an increase in the risk of PTB.

## 2. Materials and Methods

### 2.1. Study Design and Data Management

The study was designed as a retrospective cohort trial. In the first step, birth data from the “Geschäftsstelle Qualitätssicherung Rheinland Pfalz (SQMed RLP)”, which manages the data of all births taking place in the German federal state Rhineland-Palatinate (about 38,000 births per year), were requested. From this dataset, the data of all affected women were extracted, who were hospitalized in 2016 due to a threatening PTB and administered corticosteroids. All women received ACS for threatened PTB before 34 weeks of gestation. All births before the 37th week of pregnancy were defined as PTBs. This overall population was classified into eight subgroups of diagnosis (PPROM, placental bleeding, premature labor, preeclampsia, oligohydramnios, AIS, cervical insufficiency, and unknown referral diagnosis) and the risk profile was determined in relation to the described risks above. The included cases were completely anonymized patient data that are routinely collected according to the legal requirements in Germany. Beyond that, no other patients or patients’ data were included in the study.

### 2.2. Population

In total, *n* = 1544 patients were included who had been hospitalized due to a threatened PTB. All of them had been administered corticosteroids in 2016 due to a threatened PTB. All patients without symptoms of a threatened PTB were excluded.

### 2.3. Statistics

A frequency analysis was performed to determine how often individual diagnoses and risk factors were distributed throughout the collective, and how often a PTB actually occurred. The distribution of individual risk factors in the respective diagnosis groups was then calculated. To calculate the statistical significance, the Friedman test and the Chi square test were used. Logistic regression was then utilized to calculate whether patients had different odds ratios (ORs) for actual PTB with their diagnoses and identified risk factors. The respective control group always represented the whole collective without the investigated risk factor or diagnosis. Logistic regression was used to calculate whether patients with a fixed diagnosis differ with regard to the PTB risk from patients without this diagnosis and whether statistically significant correlations exist between the number of diagnoses and the risk of PTB (Spearman correlation). In order to analyze whether the risk factors influence the PTB risk of patients as a function of their diagnosis, a moderation analysis according to Hayes was performed. To identify specific risk profiles, it was subsequently determined whether risk factors increase the risk for one of the diagnoses.

Finally, the risk of PTB of the whole collective (=patients at risk of PTB) as well as the risk in relation to the respective diagnosis and the individual risk factors was compared with the risk of the entire birth cohort in Germany [1]. For the specification of the risk collectives (association between diagnosis and risk factor), a separate regression analysis was performed for each risk factor with all diagnoses. For the overall analysis in the model, the Bonferroni correction of the *p*-value was calculated in order to counteract the risk of alpha error accumulation in multiple testing. The *p*-value was divided by 8. For the comparison with the birth cohort in Germany, a separate analysis was carried out for each diagnostic group and risk factor. Women without a PTB event were considered as a confounding variable in each case.

In all analyses, *p*-values < 0.05 (two-tailed) were considered indicative of statistically significant differences (α = 0.05). All statistical analyses were conducted using IBM SPSS Statistics (version 24).

## 3. Results

### 3.1. Risk Factors and Diagnosis in the Overall Population

Table 1 shows the distribution of sociodemographic and risk factors in the overall population. The average age was 30.3 years, while 18.5% of mothers fulfilled the risk factor of being older than 35 years. Overall, 31.2% had a history of PTB, stillbirth, or abortion in their personal anamnesis, 14.7% had a multiple pregnancy, 1.4% had gestational diabetes, and 5% had hypertension, while 9.1% smoked during pregnancy. Nearly 60% of all patients who had received fetal lung maturation induction suffered an actual PTB. The other 40% of patients diagnosed with “threatened PTB” received fetal lung maturation induction without being affected by PTB. Drug induction of fetal lung maturation was most frequently performed based on the diagnosis PPROM (32.5%). Other frequent diagnoses were premature labor (26.4%) and cervical insufficiency (16%). All other diagnoses (placental bleeding: 8.0%, pre-eclampsia: 8.4%, oligohydramnios: 8.7%, and AIS: 3.9%) played a minor role. There was also the possibility of multiple diagnosis, while 22.5% could not be assigned to any of the above-mentioned diagnosis groups and were accordingly subsumed into the “none” group (Figure 1).

### 3.2. Risk of PTB in Relation to Diagnosis and Risk Factors

In total, 925 (59.9%) patients with the diagnosis “threatened PTB” suffered a PTB. The risk of PTB was highly dependent on the diagnosis groups: There was a significantly increased risk of PTB with preeclampsia (OR = 5.46, 95%-CI: 2.95–10.10), PPROM (OR = 3.23, 95%-CI: 2.51–4.16), placental bleeding (OR = 1.88, 95%-CI: 1.24–2.84), and oligohydramnios (OR = 1.75, 95%-CI: 1.14–2.68). In contrast, there was no increased but decreased risk in the diagnosis groups of premature labor (OR = 0.69, 95%-CI: 0.54–0.90) or cervical insufficiency (OR = 0.46, 95%-CI: 0.34–0.63). AIS patients had the highest risk of PTB with OR = 16.12 (95%-CI: 4.91–52.91). When considering the risk factors, multiple pregnancies (OR = 6.45, 95%-CI: 4.20–9.91) could be identified as a relevant factor for actual PTB without considering the diagnosis groups. The risk factors of age > 35 years, history of PTB, stillbirth, or abortion in the personal anamnesis, gestational diabetes, hypertension, and nicotine consumption did not model the risk of an actual PTB (logistic regression and proportion of ACS patients with the respective diagnosis or risk factor who experienced a PTB event, see Table 2).

Since the various diagnosis groups differed significantly in terms of the risk of actual PTB, it was examined whether the risk of PTB increases with the number of diagnoses. As shown in Table 3, 49.9% of the patients were assigned to one of the classified diagnosis groups, 27.6% to more than one of the diagnosis groups, and 22.5% to none of the defined diagnosis groups. There was a statistically significant correlation between the number of diagnosis groups per patient and the probability of PTB (*r* = 0.11, *p* ≤ 0.0001). Patients with a higher number of listed diagnoses had an increased risk of PTB. Compared to the diagnosis group “none”, all other diagnosis groups showed a higher risk of PTB (OR = 1.64, 95%-CI: 1.29–2.08, *p* ≤ 0.0001).

Although only multiple pregnancy was identified as a significant risk factor for PTB, a moderation analysis was performed for all risk factors (Table 4).

Since only 60 percent of women diagnosed with “threatened PTB” have experienced a PTB event, it is of interest which risk factors actually influence the PTB rate in the diagnosis groups. The risk factor of having PTB, stillbirth, or abortion in the personal anamnesis influenced the risk of PTB by 42%, depending on the diagnosis group. In contrast, the risk factor multiple pregnancy influenced the risk independently of the diagnostic group. The other risk factors mentioned above showed no significant moderation effects.

### 3.3. Risk Profile of Diagnosis Groups

The combination of diagnostic groups and risk factors allowed the identification of risk profiles (Table 5) that give a more accurate assessment of actual PTB. Furthermore, Table 5 shows the distribution of risk factors in the individual diagnosis groups that cause drug induction of fetal lung maturation. In all diagnosis groups, the risk factors of maternal age > 35 and history of PTB, stillbirth, or abortion in the personal anamnesis dominated. The differences between the diagnostic groups were statistically significant (Friedman test, *p* < 0.0001). Patients aged >35 (OR = 2.29, 95%-CI: 1.37–3.82) had an increased risk of PTB due to preeclampsia. In addition, patients with hypertension also develop oligohydramnios more frequently (OR = 2.67, 95%-CI: 1.46–4.86), which are also more frequently diagnosed in smoking pregnant women (OR = 2.99, 95%-CI: 1.79; 4.98).

### 3.4. Risk of PTB in Corticosteroid Patients in Relation to All Pregnancies

Women who have received drug induction of fetal lung maturation due to the diagnosis “threatened PTB” in Rhineland-Palatinate have a 6.6-fold increased risk of PTB compared to all pregnancies in Germany [1]. With regard to the diagnosis groups and the risk factors, the risk is highly variable (Table 6). The risk of PTB is more than 10 times higher in the diagnosis group AIS (OR = 10.61, 95%-CI: 7.51–14.98), but only 4.73 (95%-CI: 3.79; 5.90) times higher in the risk group cervical insufficiency. The optimal time for the drug induction of fetal lung maturation should therefore only be estimated with knowledge of these dependencies.

## 4. Discussion

### 4.1. Principal Results

In the present paper, an analysis of a German federal state’s pregnancy cohort emphasizes that the most frequently diagnosed indication for corticosteroid administration is PPROM followed by premature labor. Surprisingly, nearly one quarter of corticosteroid administration patients have no diagnosis at all. This is probably due to the fact that the documentation was not always complete and is a problem inherent in retrospective studies.

Nearly 60% of all patients treated with corticosteroids due to an increased risk of PTB did subsequently experience a PTB. Overall, patients who were treated with corticosteroids showed an almost seven times higher risk of PTB than women of the entire birth cohort in Germany. This is particularly important since specific risk profiles for individual diagnoses exist with regard to individual risk factors. In all diagnostic groups, the risk factor history of PTB, stillbirth, or abortion in the personal anamnesis was found particularly often, which influences the PTB risk as a moderation effect in all subgroups. In addition, there are significant differences within the risk profiles regarding the probability of a PTB event. In particular, the highest preterm risk could be found in the diagnostic groups AIS or preeclampsia as well as the risk factor of multiple pregnancies.

### 4.2. Comparison with Prior Work

Although it is well known to what extent several causes such as PPROM increase the risk of PTB, no systematic distinction has been made between patients at increased risk of PTB and those with a PTB event. In this study, it was found that various diagnoses that indicate categorization into a risk group for PTB differ greatly in terms of their actual risk of a PTB event. Studies that compare the risk of different diagnoses within the framework of the individual risk profile are absent in the literature. The proportion of pregnant women with PPROM within the risk of PTB corresponded roughly to the incidence described in the literature (25% of PTBs [14] and 3% of total births [15]) and the frequency distribution of other risk factors or diagnoses [16]. However, there are differences with regard to the incidence of PTBs; the literature indicates a significantly higher incidence of PTBs in PPROM patients (95%) [17], but a comparably high risk of PTB in patients with cervical insufficiency (40%) [18]. The difference of 95% vs. 73% in PPROM patients is possibly due to the fact that only pregnancies with lung maturity were included in the analysis. The high incidence of PTBs [1] makes the identification of risk factors and causes as well as their interaction highly relevant [7,8,9], especially with regard to the fact that risk patients for PTB have been included to decide on the indication for applying ACS [19]. It is also known from other studies that pregnant women over the age of 35 have an increased risk of PTB [20,21], although no distinction is made between patients at increased risk and those at risk of PTB.

The present study supports the risk assessment of which indications have a very high probability of a PTB event and which diagnoses are poor predictors. This may also contribute to a more target group-specific ACS prescription, the standardized administration of which in women at risk of PTB before the 24th week should be critically considered due to possible side effects [12]. Moreover, ACS is known to be most effective if it is administered within a short time frame before delivery; it should be given at least 24 h, but not more than 7 days before birth [22,23]. In another study with the same patient population, we could demonstrate that this ideal time frame was reached in only 15.2% of all pregnant women who were treated with ACS because of an increased risk of PTB (n = 1544). The ideal time frame after ACS administration was reached in less than 25% of all cases in each subgroup [24].

### 4.3. Limitations

Despite the highly significant results of this study, the analysis must be critically reflected. For the determination of the risk profiles, only women who had received an induction of fetal lung maturation were included. The total of n = 1544 patients identified represented 4.12% of the number of births in Rhineland-Palatinate in 2016 (37,518 births) [25]. It should be noted that fetal lung maturation is usually not induced by medication after 34 weeks of pregnancy. For this reason, the available results are not representative for the total collective of preterm infants, but only of the collective of PTBs < 34 + 0 weeks of gestation with the induction of lung maturation in Rhineland-Palatinate. There were n = 925 PTBs after the drug induction of fetal lung maturation in Rhineland-Palatinate in 2016, which corresponds to 2.47% of all births in this state. Only 1.4% of the women examined were diagnosed with gestational diabetes, which underlines that this risk factor does not represent a modulation of the risk of PTB. It is also conceivable that this parameter was insufficiently documented (as it was not a mandatory field in the documentation); other studies show a significantly higher prevalence of gestational diabetes in PTB [26]. It should also be noted that unfortunately no information was available on the ACS dose. Patients are normally given a weekly dose. In the present study, all patients who received at least one ACS dose due to a threatened PTB were included. The number of doses administered per patient was not documented. A further limitation resulted from the fact that only the preterm infants of one cohort were available. This was due to the fact that the data were not read out by our working group, but were made available as part of standardized quality monitoring. A better representativeness could have been achieved if several cohorts could have been evaluated cumulatively. Further limitations arise from the fact that the analyzed routine data of the quality monitoring do not contain any information about the exact time and criteria of diagnosis. However, we assume that the diagnoses were made before the PTB event, so they are not added retrospectively. The quality monitoring in Germany works with cumulative, successively collected data that should not be modified retrospectively. This also explains the high proportion of patients without a diagnosis. In particular, the accurate diagnosis of PPROM is considered challenging, since there is still no consensus on the criteria to diagnose PPROM and there is very little evidence on the accurate prediction of women with PPROM that are more likely to deliver within seven days [13,27]. “No diagnosis” means that there was no diagnosis in the medical record that could be transmitted as part of the data compilation.

### 4.4. Clinical Implications

A history of PTB, stillbirth, or abortion in the personal anamnesis can be identified as a significant risk factor for PTB. The combination of our diagnosis groups and this risk factor shows the significantly higher risk of a PTB in patients after the induction of fetal lung maturation. The data show that the induction of fetal lung maturation in the absence of this risk factor must be very critically reconsidered, since actual PTB occurs in this situation much less frequently.

## 5. Conclusions

It can be observed that about 60% of patients treated with corticosteroids for the induction for fetal lung maturation also have a PTB event. The risk of PTB in these patients is 6.6 times higher than in the national cohort. Patients diagnosed with AIS, pre-eclampsia, and multiple pregnancies were found to have a particularly high risk, while risk factors prior to PTB, stillbirth, or abortion act as a relevant confounding factor. Not all women with an increased risk of PTB have a PTB event. In order to decide which patients benefit from the induction of fetal lung maturation, it is necessary to distinguish between patients who are only classified as being at risk of PTB and those with a PTB event in combination with their diagnoses.

## Figures and Tables

**Figure 1 healthcare-13-00087-f001:**
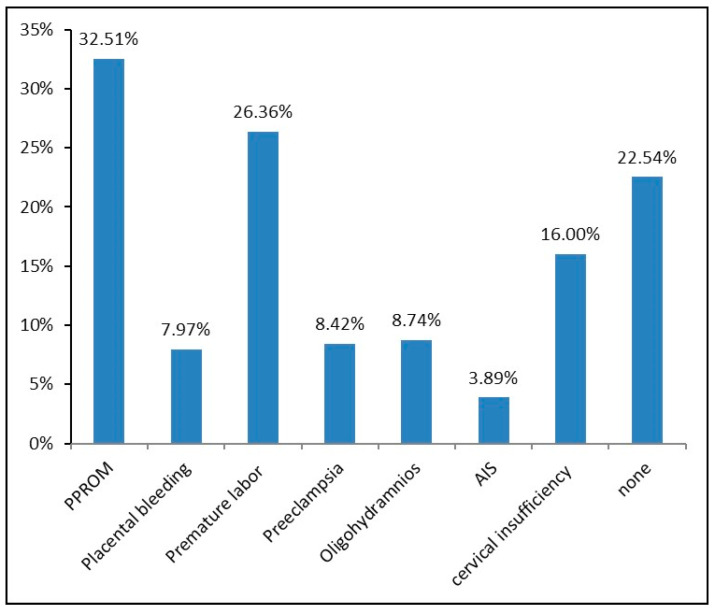
Distribution of diagnoses in the study population, responsible for the induction of lung maturation, *n* = 1544. AIS = amniotic infection syndrome; PPROM = preterm premature rupture of membranes.

**Table 1 healthcare-13-00087-t001:** Sociodemographic and risk factors in the study collective, *n* = 1544.

Factor	Value
Age (years)	
Mean (SD)	30.3 (5.61)
Median (Min; Max)	31 (14; 48)
Age > 35	285 (18.5%)
History of PTB, stillbirth, or abortion in personal anamnesis	481 (31.2%)
Multiple pregnancies	227 (14.7%)
Gestational diabetes	22 (1.4%)
Hypertension	77 (5.0%)
Nicotine abuse	140 (9.1%)
PTB	925 (59.9%)

**Table 2 healthcare-13-00087-t002:** Odds ratio for PTB in the study population separated in relation to diagnosis groups and risk factors, *n* = 1544.

Diagnosis Groups				
	%	OR	95%-CI	*p*-Value
PPROM	73.6%	3.23	[2.51; 4.16]	<0.0001 *
Placental bleeding	68.9%	1.88	[1.24; 2.84]	0.003 *
Premature labour	48.0%	0.69	[0.54; 0.90]	0.005 *
Preeclampsia	87.5%	5.46	[2.95; 10.10]	<0.0001 *
Oligohydramnios	71.0%	1.75	[1.14; 2.68]	0.010 *
AIS	95.5%	16.12	[4.91; 52.91]	<0.0001 *
Cervical insufficiency	42.6%	0.46	[0.34; 0.63]	<0.0001 *
**Risk factors**				
Age mother > 35	66.7%	1.08	[0.79; 1.47]	0.636
History of PTB, stillbirth, or abortion in personal anamnesis	55.1%	0.79	[0.611; 1.01]	0.062
Multiple pregnancies	87.7%	6.45	[4.20; 9.91]	<0.0001 *
Gestational diabetes	40.9%	0.46	[0.18; 1.21]	0.116
Hypertension	83.1%	1.61	[0.76; 3.39]	0.210
Nicotine abuse	55.7%	1.04	[0.70; 1.53]	0.857

* Statistically significant at *p* < 0.05, logistic regression, the respective control group always represented the whole collective without the investigated risk factor or diagnosis. In all models, ACS patients without a PTB event were controlled as confounders. Patients with multiple diagnoses were considered separately for each diagnosis/risk factor. AIS = amniotic infection syndrome; CI = confidence interval; OR = Odds Ratio; PPROM = preterm premature rupture of membranes; % = Number of ACS patients with the respective diagnosis or risk factor who experienced a PTB event.

**Table 3 healthcare-13-00087-t003:** Proportion of patients with 1 diagnosis, >1 diagnosis, or no diagnosis and risk of PTB in relation to number of diagnoses, *n* = 1544.

	*n*	%	r	OR	95%-CI	*p*-Value
1 of the diagnoses in Table 1	770	49.9%				
2 of the diagnoses in Table 1	350	22.7%				
3 of the diagnoses in Table 1	67	4.3%				
4 of the diagnoses in Table 1	9	0.6%				
None of the diagnoses in Table 1	348	22.5%				
Correlation analyses: number of diagnoses and risk of PTB			0.11		[0.063; 0.158]	<0.0001 *
OR diagnosis				1.64	[1.29; 2.08]	<0.0001 *

* Statistically significant at *p* < 0.05; *r* = correlation coefficient (Spearman Rho); OR was calculated with logistic regression. In all models, ACS patients without a PTB event were controlled as confounders. Patients with multiple diagnoses were considered separately for each diagnosis/risk factor. CI = confidence interval; OR = Odds Ratio.

**Table 4 healthcare-13-00087-t004:** Moderation effect of risk factors to risk of PTB, *n* = 1544.

Risk of PTB: Moderation Effect of… to Diagnosis	Z-Score	SE	Indirect Effect of X of Y *	95%-CI	*p*-Value
Maternal age > 35	−0.74	0.23	−0.17	[−0.63; 0.29]	0.46
History of PTB, stillbirth, or abortion in personal anamnesis	−2.25	0.18	−0.42	[−0.78; −0.05]	0.025 *
Multiple pregnancies	−3.34	0.39	−1.31	[−2.08; −0.54]	0.0008 *
Gestational diabetes	0.99	0.77	0.76	[−0.75; 2.28]	0.324
Hypertension	0.99	0.78	0.76	[−0.76; 2.13]	0.35
Nicotine abuse	0.73	0.40	−0.48	[−0.50; 1.08]	0.47

* Statistically significant at *p* < 0.05; moderation analysis according to Hayes. In all models, ACS patients without a PTB event were controlled as confounders. Patients with multiple diagnoses were considered separately for each diagnosis/risk factor. CI = confidence interval; SE = standard error.

**Table 5 healthcare-13-00087-t005:** Identification of risk profiles in relation to the respective diagnosis.

		PPROM	Placental Bleeding	Pre-Mature Labour	Pre-Eclampsia	Oligo-Hydramnios	AIS	Cervical In-Sufficiency	None
**Age mother > 35**	**%**	20.3%	13.8%	18.9%	20.0%	15.6%	20.0%	24.7%	19.0%
	**OR**	1.02	1.15	0.63	2.29	1.07	1.17	1.44	0.81
	**95%-CI**	[0.77; 1.34]	[0.73; 1.82]	[0.45; 0.88]	[1.37; 3.82]	[0.68; 1.70]	[0.63; 2.18]	[1.04; 2.0]	[0.55; 1.18]
	** *p* **	0.91	0.54	0.0061 *	0.002 *	0.78	0.62	0.03	0.27
**History of PTB, stillbirth, or abortion in personal anamnesis**	**%**	33.5%	35.0%	32.4%	31.5%	35.6%	45.0%	33.2%	31.6%
	**OR**	0.93	1.22	1.04	0.49	0.74	1.31	0.95	0.85
	**95%-CI**	[0.74; 1.18]	[0.83; 1.80]	[0.80; 1.37]	[0.28; 0.85]	[0.49; 1.12]	[0.76; 2.23]	[0.71; 1.27]	[0.62; 1.16]
	** *p* **	0.56	0.31	0.75	0.01	0.16	0.33	0.72	0.31
**Multiple pregnancy**	**%**	15.5%	15.4%	15.0%	16.2%	8.2%	18.3%	15.4%	15.5%
	**OR**	0.81	0.78	0.51	1.34	1.44	1.70	0.95	0.66
	**95%-CI**	[0.60; 1.10]	[0.43; 1.43]	[0.34; 0.77]	[0.73; 2.63]	[0.86; 2.42]	[0.88; 3.29]	[0,65; 1.39]	[0.41; 1.05]
	** *p* **	0.18	0.43	0.001 *	0.32	0.17	0.12	0.79	0.079
**Gestational diabetes**	**%**	1.4%	2.4%	2.0%	1.5%	1.5%	3.3%	2.0%	0.3%
	**OR**	0.66	0.45	0.37	2.91	1.81	-	1.36	1.43
	**95%-CI**	[0.26; 1.69]	[0.59; 3.46]	[0.10; 1.29]	[0.80; 10.62]	[0.50; 6.49]	-	[0.49; 3.73]	[0.50; 4.11]
	** *p* **	0.38	0.45	0.12	0.11	0.37	-	0.55	0.51
**Hypertension**	**%**	4.2%	3.3%	5.4%	10.0%	5.2%	3.3%	4.5%	4.6%
	**OR**	0.17	0.57	0.21	109.57	2.67	0.24	0.77	0.20
	**95%-CI**	[0.08; 0.37]	[0.23; 1.46]	[0.09; 0.45]	[50.91; 235.84]	[1.46; 4.86]	[0.03; 1.78]	[0.40; 1,49]	[0.07; 0.55]
	** *p* **	<0.0001 *	0.24	<0.0001 *	<0.0001 *	0.001 *	0.16	0.44	0.002 *
**Nicotine abuse**	**%**	9.4%	11.4%	13.0%	9.2%	8.9%	8.3%	8.9%	8.9%
	**OR**	1.22	1.12	1.00	1.05	2.99	0.76	1.10	0.87
	**95%-CI**	[0.67; 2.23]	[0.62; 2.02]	[0.65; 1.53]	[0.44; 2.51]	[1.79; 4.98]	[0.30; 1.93]	[0.46; 2.64]	[0.52; 1.44]
	** *p* **	0.51	0.71	0.99	0.91	<0.0001 *	0.84	0.67	0.58

* Statistically significant at *p* ≤ 0.00625 (after Bonferroni correction), logistic regression, the respective control group always represented the whole collective without the investigated risk factor or diagnosis. In all models, ACS patients without a PTB event were controlled as confounders. Patients with multiple diagnoses were considered separately for each diagnosis/risk factor. AIS = amniotic infection syndrome; CI = confidence interval; OR = Odds Ratio; PPROM = preterm premature rupture of membranes; % = number of patients within the respective diagnosis group who also had the risk factor.

**Table 6 healthcare-13-00087-t006:** Risk of PTB in corticosteroid patients in comparison to the national birth cohort.

	OR	95%-CI	*p*-Value
**Diagnosis groups**			
All corticosteroid patients	6.66	[6.13; 7.22]	<0.0001 *
PPROM	8.18	[7.18; 9.32]	<0.0001 *
Placental bleeding	7.65	[5.88; 9.97]	<0.0001 *
Premature labour	5.33	[4.52; 6.30]	<0.0001 *
Preeclampsia	9.72	[7.60; 12.44]	<0.0001 *
Oligohydramnios	7.89	[6.13; 10.16]	<0.0001 *
AIS	10.61	[7.51; 14.98]	<0.0001 *
Cervical insufficiency	4.73	[3.79; 5.90]	<0.0001 *
**Risk factors**			
Age mother > 35	7.41	[6.16; 8.90]	<0.0001 *
History of PTB, stillbirth, or abortion in personal anamnesis	6.12	[5.27; 7.11]	<0.0001 *
Multiple pregnancies	9.74	[8.05; 11.78]	<0.0001 *
Gestational diabetes	4.55	[2.09; 9.87]	<0.0001 *
Hypertension	9.24	[6.63; 12.87]	<0.0001 *
Nicotine abuse	6.19	[4.69; 12.90]	<0.0001 *

* Statistically significant at *p* < 0.05, logistic regression, the respective control group always represented the whole collective without the investigated risk factor or diagnosis. In all models, ACS patients without a PTB event were controlled as confounders. Patients with multiple diagnoses were considered separately for each diagnosis/risk factor. AIS = amniotic infection syndrome; CI = confidence interval; OR = Odds Ratio; PPROM = preterm premature rupture of membranes.

## Data Availability

The data that support the findings of this study are available from the corresponding author upon reasonable request.

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
