# Peer review of "Likelihood of Preterm Birth in Patients After Antenatal Corticosteroid Administration in Relation to Diagnosis and Confounding Risk Factors: A Retrospective Cohort Study"

_healthcare, 2025, doi:10.3390/healthcare13010087_

Round 1
Reviewer 1 Report
Comments and Suggestions for Authors
I have read this paper with a background on perinatal clinical research and clinical pharmacology. The paper is well timed, as there are increasing concerns on outcomes in fetusses inadverted exposure to prenatal lung maturation (inadverted defined as subsequent born at term equivalent age). Perhaps this inadverted outcome should be further stressed, as eg summarized in the WPAM clinical practice guidelines on antenatal corticosteroids for fetal maturation.
Since the initial papers on lung maturation, about 50 % of women subsequently does not deliver, so that additional research is needed in this field (in this way, 60 % is not ‘great’, but still reasonable). This paper explores the association between diagnosis and preterm delivery in this ‘a priori’ perceived high risk group.
Related to this, it is perhaps useful to reconsider the word ‘risk’, and use ‘Likelihood’ as alternative in the title ?
To ensure accurate interpretation and related to the retrospective nature, there should be clarity on how and when diagnoses were made. To illustrate this, AIS prospective before delivery is likely different from post hoc, APO based diagnosis. Are you aware of eg preeclampsia diagnosis <34 wks that did not receive prenatal steroids (I assume that this is an unlikely scenario, but a limitation of the current design ? This is a present (time of diagnostic allocation versus administration of prenatal steroids). Another weakness of the retrospective analysis (figure 1) si ‘none’, likely no diagnosis retrieved in the medical file ?
For clarify, preterm delivery was any delivery < 37 weeks in your analysis ?
Reviewer 2 Report
Comments and Suggestions for Authors
-As a reviewer, my suggestion would be not to separate the introduction section. Instead, I recommend combining it into a cohesive narrative and including a clear explanation of the term "immune preterm birth," as it is not currently indexed in MeSH. This would help improve clarity and accessibility for readers unfamiliar with the term.
-As a reviewer, I recommend revising the sentence in the introduction section that begins with "N = 1,544..." to ensure it does not start with a numeral. Rephrasing the sentence to provide context before introducing the figure would improve readability and maintain a more formal tone. For example, you could say: "The study analyzed a total of 1,544 cases, demonstrating..." or a similar construction.
- In the text, the letter "n" is inconsistently capitalized—appearing as both uppercase and lowercase. I recommend standardizing this usage throughout the manuscript for consistency.
-In the text, terms like "premature" and "preterm birth/delivery" are used interchangeably, leading to inconsistency. Additionally, the term "immune preterm birth" is not consistently abbreviated, and its definition is not clearly provided. Furthermore, the population associated with "immune preterm birth" is not adequately described. I recommend clarifying and standardizing these terms throughout the manuscript for better readability and comprehension.
-The keywords used in the manuscript are not listed in MeSH. I recommend revising the keywords to align with MeSH terms to ensure better indexing and discoverability in medical databases.
-The tables in the Results section require formatting improvements for better clarity and presentation.
-In the Results section, I recommend specifying the measure of dispersion used alongside the Odds Ratio in the text.
-The footers for Table 2 is in the same font and style as the main text, making it difficult to distinguish. I recommend formatting differently to enhance clarity.
-Could you please clarify the difference between the results already reported in this article (https://doi.org/10.1002/ijgo.15052) and the findings presented in this manuscript? This would help highlight any novel contributions and ensure a clearer distinction between the two studies.
Round 2
Reviewer 2 Report
Comments and Suggestions for Authors
Ensure consistent use of abbreviations throughout the text. For instance, PPROM is referred to as PROM in some sections, which may cause confusion. Additionally, avoid repeatedly explaining the same abbreviations; once defined, subsequent mentions should simply use the abbreviation.
Author Response
Dear Reviewer,
Dear Reviewer,
Thank you for your critical review of our paper. We found one “PROM” label in the text via the search-and-replace function in Word and replaced it with “PPROM”.
Rereading the paper revealed that all abbreviations are only explained once in the main text. However, the journal's guidelines state that abbreviations must also be explained separately in the abstract (because usually only the abstracts are read after a PubMed search) and all tables must be self-explanatory, which includes that each abbreviation must be resolved again under each table.
Should there be any redundancies (of which we have not identified any at this point), these will be reliably removed by the journal's copy editor after acceptance.
With best regards